# Contribution of social determinants to symptoms of generalized anxiety disorder

**Jerzy Bala, Jennifer J. Newson⬤, Tara C. Thiagarajan⬤***

Sapien Labs, Wilson Blvd, floor, Arlington, Virginia, United States of America

* tara@sapienlabs.org

## Abstract

Symptoms of anxiety are known to be triggered by a range of life context factors including early life trauma, poor sleep quality, infrequent exercise, unemployment and social isolation. Machine learning techniques offer a powerful method for analyzing these factors in combination, enabling the evaluation of aggregate predictive associations rather than causal pathways and the identification of their relative association with anxiety symptoms. However, most studies examining these factors have either been small-scale or included only a small number of factors. Here we applied multiple machine learning approaches (Random Forest, Gradient Boosting, Naïve Bayes, Information Gain, and SHAP) to a cross-sectional data sample of 4,186 individuals to reveal how a broad range of lifestyle and life context factors are associated with the experience of anxiety symptoms, as measured by the Generalized Anxiety Disorder-7 screening questionnaire (GAD-7). The results showed that, in combination, early life trauma, poor sleep quality, infrequent exercise, unemployment, and deterioration of social bonds were substantially associated with anxiety symptoms, particularly for older age groups, with frequency of a good night's sleep having an outsized impact. For older ages, this was followed by employment status and experience of interpersonal trauma, as well as frequency of in-person socializing. For younger ages (18–34), employment status was less important with interpersonal trauma being a more significant factor. Specifically, poor sleep, rarely socializing in person, not being able to work or being unemployed, bullying by peers, or neglect/abuse by a parent or caregiver had the largest associations with anxiety symptoms. These findings have implications for how we approach both prevention and treatment of anxiety.

## Introduction

Feelings of fear and anxiety, evoked by threats, or the anticipation of real or imagined threats, are natural adaptive responses which facilitate survival. However, when these feelings become excessive in frequency, severity or duration, they can become maladaptive and functionally impairing. Although there are multiple disorders

**Data availability statement:** The full dataset from the Global Mind Project, including the data used in this study, is freely available for not-for profit purposes from the Sapien Labs Researcher Hub. Access can be requested here: https://sapienlabs.org/global-mind-project/researcher-hub/.

**Funding:** This work was supported by funding from Sapien Labs.

**Competing interests:** The authors have declared that no competing interests exist.

associated with maladaptive anxiety (e.g., phobias, panic, social anxiety), Generalized Anxiety Disorder (GAD) is among the most prevalent and clinically significant anxiety-related conditions, and is described by the DSM-5 as 'excessive worry and apprehensive expectations, occurring more days than not for at least 6 months, about a number of events or activities, such as work or school performance' [1]. Within a primary care or therapeutic setting, it is often initially screened for using the General Anxiety Disorder-7 questionnaire (GAD-7) which includes 7 questions on worry, fear, restlessness, irritability, and where respondents rate each item according to how frequently they have been bothered by the problem over the last two weeks [2]. A sum score threshold of 10 on the GAD-7 (moderate anxiety) has been shown to have a sensitivity of 89% and a specificity of 82% for GAD, as determined by telephone interview with a mental health professional [2].

Although there are various treatments available to help reduce or manage symptoms of anxiety (e.g., cognitive behavioural therapy, psychoactive medications), its high and increasing prevalence in the general population [3–6], debilitating life impact [7,8] and low treatment availability [9], means there is a need to better understand underlying risk factors such that incidence can be preventatively reduced. However, the aetiology and risk profile of anxiety is complex and still poorly understood. Although family studies have revealed a genetic component [10,11], people's environment, life experiences and lifestyle habits (together, social determinants) all play a prominent role in the onset and trajectory of the anxiety symptoms. For example, a range of factors including early life trauma [12–14], poor sleep [15,16], unhealthy diet [17,18], infrequent exercise [19,20], unemployment [21,22] and social isolation [23,24] have all been associated with anxiety symptoms, often bidirectionally.

Machine learning techniques offer a powerful method to analyze these factors in combination to predict outcomes and identify their relative association with anxiety symptoms [see [25–29] for reviews]. However, to date, most machine learning studies in this context have either been small scale (typically a few hundred people) [30–34], only included a limited number of factors (e.g., demographics and medical history) [35,36], and/or focused on specific populations (e.g., a specific country, age group or clinical population) [31,34–46]. As a result, it is currently unknown how well anxiety symptoms can be predicted by lifestyle and life circumstance factors in the aggregate, and which ones are more important predictors of anxiety symptoms in the general population. This has implications not only for identifying at risk populations and directing health public policy, but also in terms of how anxiety symptoms can be prevented and how they should be treated from a therapeutic perspective [47–49].

Here we take advantage of a large-scale cross-sectional data collection effort that asked a subsample of respondents from the general population (N = 4,186) to complete the GAD-7 screening questionnaire, and included responses on a rich array of lifestyle and life experience factors. In this study, individuals, taken from a general population, completed the GAD-7 questionnaire and answered questions that described their lifestyle, life circumstance, and experience of various adversities and interpersonal traumas. We applied multiple machine learning approaches (Random Forest, Gradient Boosting, Naïve Bayes, Information Gain and SHAP) to this data

sample to determine the degree to which these lifestyle and life experience factors, in the aggregate, were associated with GAD-7 outcomes in terms of statistical classification of symptom severity (rather than prospective prediction), and to describe the relative importance of these different factors. These findings have implications not only for identifying at risk populations and directing health public policy, but also in terms of how anxiety can be prevented and how it should be treated from a therapeutic perspective [47–49].

## Methods

### Data acquisition and data elements

Participants were recruited as part of the ongoing Global Mind Project through online advertisements placed on Facebook and Google that targeted age-sex groups and geographical regions across broad based interests and key words [50]. The recruitment and data quality procedures have been previously described in detail [59]. Respondents were aged 18–85, predominantly from 20 countries, with 45.7% of respondents reporting their biological sex as male (see Table A in S1 Text for age and sex break-up and Table B in S1 Text for percentage of respondents by country). Participants were directed to the survey website (https://sapienlabs.org/mhq/) and completed the GAD-7 questions as part of a larger assessment of mental wellbeing [51,52]. Respondents also answered questions on a broad range of life context factors including lifestyle habits, life circumstances and experience of various adversities and interpersonal traumas (see Table 1).

Cross-sectional data was collected between 14/09/2022 and 29/09/2022, during which 4,421 respondents completed the GAD-7. Standard Global Mind Project cleaning criteria were applied to the data. Only respondents who responded 'Yes' to the MHQ question 'Did you find this assessment easy to understand?', and who had a standard deviation of >0.2 across MHQ rated question responses were included in the analysis, leading to a final sample size of 4,186. This criterion, part of the validated MHQ quality-control framework [51,59], uses within-respondent response variability to identify disengaged or inattentive respondents; SD < 0.2 across 47 MHQ items indicates extremely low variability (e.g., selecting the same response for nearly all items), a pattern empirically associated with poor data quality.

Participants took part in the online survey voluntarily, anonymously, and without any financial compensation. Participants consented to take part by clicking on a start button after reading a detailed privacy policy. Cases with missing data on predictor variables were excluded from analysis on a listwise basis. All procedures involving human subjects were approved by the Health Media Lab Institutional Review Board (HML IRB; OHRP Institutional Review Board #00001211, Federal Wide Assurance #00001102, IORG #0000850).

### Calculation of GAD-7 Sum Scores

No post-stratification weights were applied to the sample, as the study's primary purpose was model performance evaluation rather than population prevalence estimation. Each GAD-7 item was rated on a frequency scale of 0–3 that reflected how much a symptom had bothered them over the last 2 weeks (0 = *Not at all*; 1 = *Several days*; 2 = *More than half the days*; 3 = *Nearly every day*). The sum of these ratings, the GAD-7 sum score, was computed for each respondent, and the proportion of respondents within each category (*Minimal anxiety* = 0–4; *Mild anxiety* = 5–9; *Moderate anxiety* = 10–14; *Severe anxiety*=≥15) was calculated. GAD-7 sum scores in this data spanned a full range from 0 to 21 with 24.1% having scores of 10 or higher (Fig 1).

### Classification models

Multiple classification models were used to identify the model type with the best performance. These included Logistic Regression as well as tree-based models such as Random Forest, Gradient Boosting (XGBoost) and Naïve Bayes using Orange Data Mining, an open-source machine learning and data visualization toolkit designed for data analysis through

**Table 1. Life context factors queried in the survey.**

| Group | Factor |
|---|---|
| Demographics | Age; Biological Sex; Employment status; Educational attainment; |
| Geography | Country, State; City; |
| Lifestyle | Frequency of getting a good night's sleep; Frequency of doing exercise; Frequency of Socializing |
| Substance use | Tobacco products (e.g., cigarettes, chewing tobacco, cigars, etc.); Vaping products; Alcoholic beverages (e.g., beer, wine, spirits, etc.); Cannabis (e.g., marijuana, pot, grass, hash, etc.); Cocaine (e.g., coke, crack, etc.); Amphetamine type stimulants (e.g., speed, diet pills, ecstasy, etc.); Inhalants (e.g., nitrous, glue, petrol, paint thinner, etc.); Sedatives or Sleeping Pills (e.g., Benzodiazepines, Valium, Serepax, Rohypnol, etc.); Hallucinogens (e.g., LSD, acid, mushrooms, PCP, Special K, etc.); Opioids (e.g., heroin, morphine, methadone, codeine, etc.); Barbiturates (e.g., Nembutal, Pentobarbital) |
| Medical | Presence/absence of diagnosed medical disorder; Type of medical condition; Mental health treatment status; Reasons for not seeking help; Type of mental health treatment; Diagnosed mental health disorder(s) |
| Childhood Inter-personal traumas | Parental Divorce or family breakup; Prolonged physical abuse, or severe physical assault; Prolonged sexual abuse, or severe sexual assault; Physical violence in the home between family members (e.g., between parents); Cyberbullying or online abuse; Prolonged or sustained bullying in person from peers; Prolonged emotional or psychological abuse or neglect from parent/caregiver; Lived with a parent/caregiver who was an alcoholic or who regularly used street drugs; Threatening, coercive or controlling behavior by another person; Forced family control over major life decisions (e.g., marriage); Parent/Caregiver/Sibling with mental illness or who committed suicide; Parent/Caregiver/Sibling went to prison; Serious injury, harm, or death you caused to someone else |
| Childhood Financial Adversities | Extreme poverty leading to homelessness and/or hunger |
| Childhood Other Adversities | Life threatening or debilitating injury or illness; Sudden or premature death of a parent or sibling; Involvement or close witness to a war; Displacement from your home due to political, environmental or economic reasons; Suffered a loss in a major fire, flood, earthquake, or natural disaster; Caring for a parent or sibling with a major chronic disability or illness |
| Adulthood Interpersonal Traumas | Divorce/separation or family breakup; Prolonged physical abuse, or severe physical assault; Prolonged sexual abuse, or severe sexual assault; Cyberbullying or online abuse; Serious injury, harm, or death you caused to someone else; Threatening, coercive or controlling behavior by another person; Forced family control over major life decisions (e.g., marriage) |
| Adulthood Financial Adversities | Extreme poverty leading to homelessness and/or hunger; Loss of your job or livelihood leading to an inability to make ends meet |
| Adulthood Other Life Adversities | Life threatening or debilitating injury or illness; Sudden or premature death of a loved one; Caring for a child or partner with a major chronic disability or illness; Involvement or close witness to a war; Suffered a loss in a major fire, flood, earthquake, or natural disaster; Displacement from your home due to political, environmental or economic reasons |

visual programming or Python scripting (https://orangedatamining.com/). The Logistic Regression model was implemented with L1 (LASSO) regularization (the cost strength C = 12) and class balancing enabled. Table C in S1 Text provides the full model comparison across all classifiers. Models were created for the identification of individuals with GAD-7 scores ≥10 and <10 separately, then combined into a composite Logistic Regression model. This approach trains separate classifiers on minority and majority classes to mitigate imbalanced-data bias and is an overfitting-mitigation strategy rather than a method for imputing missing outcome data. All features were one-hot encoded where each answer option, if selected, was coded as a 1 and if not selected, as a 0.

Performance metrics including ROC area under the curve (AUC), accuracy, precision, recall and F1 scores were computed. This was done using all the data together, as well as separating the data by both geography and age. Geographically, models were built separately for all data acquired from western/developed countries (N = 7 countries; 40% of sample) and from non-western/developing countries (N = 13 countries; 60% of sample). Similarly, models were built for each decadal age group, pooling all geographies. Results reported are based on a 5-fold stratified cross-validation.

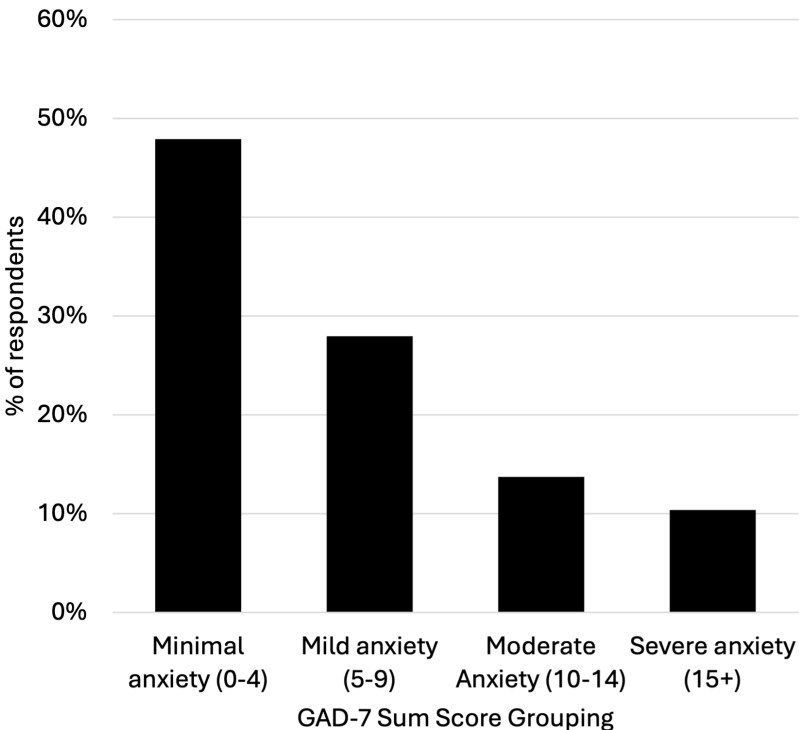

**Fig 1. Histogram of GAD-7 sum scores in the sample.**

### Estimation of contribution of variable categories to model performance

Using Logistic Regression models, the contribution of each category of lifestyle or life experience factor to the model performance in predicting symptoms of moderate to severe anxiety, defined as GAD-7 scores ≥10, was evaluated. These included frequency of exercise, frequency of socializing, employment status, number of interpersonal traumas, number of financial adversities, number of other life adversities and substance use. For each feature category (e.g., all exercise frequencies for the frequency of exercise question), the increase in each performance metric was evaluated at different positions of forward addition, when added first to a base model that included biological sex, and also when added last after all other feature categories/features had been included.

### Information gain

Information Gain was used to assess the contribution of each feature (i.e., one hot encoded option) computed as the reduction in binary entropy of a target variable when it is split based on a particular feature [53]. For feature categories, aggregate information gain values were computed by averaging the individual Information Gain values for each feature/answer option within the category.

### SHAP

We used the SHAP method to compute Shapley values, to assess how specific features affected prediction outcomes. Briefly, the marginal contribution of a feature (a one-hot encoded option of a factor such as exercise) was computed for each grouping as the difference in the predicted outcome with and without the feature. The Shapley value was the

(weighted) average of marginal contributions, providing a view of both the magnitude and direction of each feature's contribution [54].

## Results

### Prediction of moderate to severe anxiety by lifestyle and life circumstance

Here we used multiple models (Logistic Regression, Random Forest, Gradient Boosting and Naïve Bayes), to determine how well, in aggregate, multiple lifestyle and life experience factors predicted symptoms of moderate to severe anxiety, defined as a GAD-7 score of 10 or higher. Given that only 24% of the sample reported symptoms moderate to severe anxiety, models that classified high GAD-7 scores tended to overfit in the training while models that predicted the converse, tended to over generalize. The tree-based models, including Gradient Boosting and Random Forest, tended to overfit the majority class (GAD-7 < 10; Table C in S1 Text). This is evidenced by their high precision (0.82 and 0.86, respectively) and F1 scores (0.88 and 0.87, respectively) for this class, but considerably lower performance on the minority class (GAD-7 > 10), with lower recall (0.25 and 0.47, respectively) and F1 scores (0.33 and 0.49, respectively). When faced with imbalanced datasets (i.e., unequal class distributions, where only 24% of the sample had GAD-7 scores ≥10), this overfitting to the majority class is characteristic of decision tree-based algorithms, as they tend to create overly complex models that capture noise in the majority class. Naive Bayes showed similar, though slightly poorer performance compared to tree-based models. Logistic Regression, while not achieving the absolute highest scores in any single metric, demonstrated the most balanced performance across both classes, as evidenced by its superior AUC (0.80) and F1-score (0.53) for the GAD-7 > 10 class. This balanced performance occurs because tree-based models often maximize overall accuracy by correctly classifying the majority class at the expense of minority class detection, whereas Logistic Regression's linear decision boundary provides more stable probability estimates across classes, making it the most suitable model for this study's objective of identifying individuals with moderate to severe anxiety symptoms. Aggregate performance was similar even when models were created separately for western developed and non-western developing countries (Table 2, bottom rows), with AUC scores of 0.81 and 0.77 and F1-scores of 0.76 and 0.74, respectively. Therefore, Logistic Regression models combining all geographic regions (i.e., both western/developed and non-western/developing countries) were used for further analysis.

However, model performance, indicating the ability to classify moderate to severe anxiety symptoms based on lifestyle and life experience factors, was systematically and significantly lower for younger age groups relative to older age groups (Fig 2, Table 3).

Accuracy and F1 scores were 0.77 and 0.86, respectively, for those age 65 and older but only 0.67 and 0.68 for those under age 34, while performance was in between for the middle age groups. This suggests that while older age groups were more likely to experience anxiety symptoms associated with the lifestyle and adverse life circumstances captured here, younger age groups were increasingly likely to experience anxiety symptoms associated with other factors not included in this model.

### Hierarchy of factors contributing to model performance

Many factors contributing to anxiety may be inter-related. For example, one might sleep worse if not exercising or if experiencing interpersonal trauma, or one might be more likely to use substances such as tobacco or alcohol when

**Table 2. Composite Logistic Regression performance for classification of moderate to severe anxiety symptoms.**

| Composite Model | AUC | Accuracy | Precision | Recall | F1 |
|---|---|---|---|---|---|
| All Countries | 0.80 | 0.73 | 0.80 | 0.73 | 0.75 |
| Western Developed Countries | 0.81 | 0.75 | 0.81 | 0.75 | 0.76 |
| Non-Western Developing Countries | 0.77 | 0.72 | 0.79 | 0.72 | 0.74 |

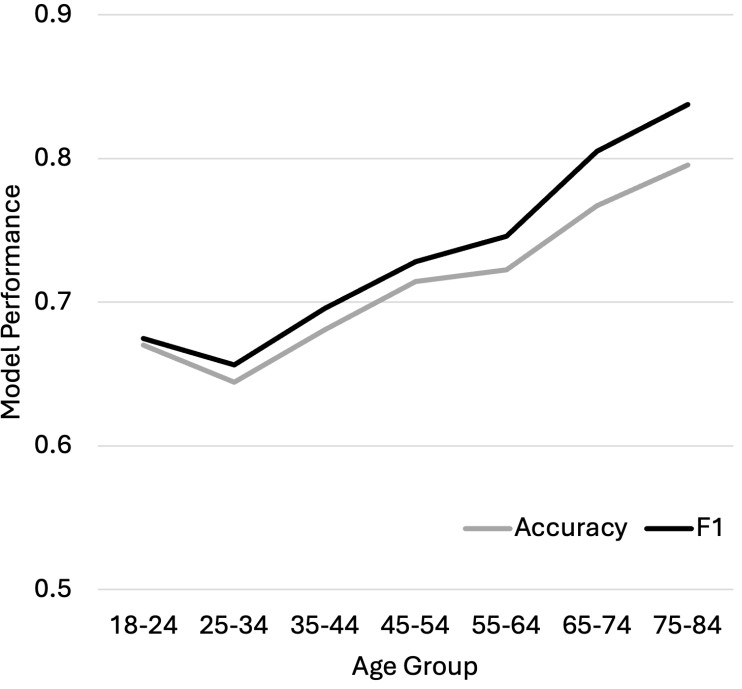

**Fig 2. Accuracy and F1 scores for the classification of moderate to severe anxiety symptoms using Logistic Regression showed better performance for older age groups.**

**Table 3. Logistic regression model performance by age group.**

| Age Group | AUC | Accuracy | Precision | Recall | F1 |
|---|---|---|---|---|---|
| 18-24 | 0.69 | 0.67 | 0.68 | 0.67 | 0.67 |
| 25-34 | 0.65 | 0.64 | 0.68 | 0.64 | 0.66 |
| 35-44 | 0.72 | 0.68 | 0.73 | 0.68 | 0.70 |
| 45-54 | 0.73 | 0.71 | 0.76 | 0.71 | 0.73 |
| 55-64 | 0.78 | 0.72 | 0.79 | 0.72 | 0.75 |
| 65-74 | 0.78 | 0.77 | 0.88 | 0.77 | 0.81 |
| 75-84 | 0.61 | 0.80 | 0.90 | 0.80 | 0.84 |

unemployed. We therefore examined the impact of adding lifestyle and life experiences factors on model performance (AUC and F1 scores) when they were added either first or last (Fig 3A, Table 4, Table D in S1 Text When added first this indicates the contribution of the factor inclusive of its interactions and correlations with other factors, while adding it last provides insight into the contribution of the factor independent of its interactions and correlations with other factors.

We performed this both for all ages together and for the 18–34 age group separately. Contributions of all factors diminished substantially when added last compared to when added first due to inter-relationships between factors for all ages and for the18–34 age group alone. For all age groups together, frequency of good sleep contributed the most to both AUC and F1 scores (0.163 and 0.041, respectively), while employment status had the second highest impact (0.108 and 0.017, respectively) followed by the experience of interpersonal trauma, frequency of social interaction and frequency of exercise. Educational attainment, substance use, financial adversities, and other adversities ranked lower, with no impact when added last. Similarly, frequency of good sleep also ranked highest in its contribution to AUC and F1 scores overall

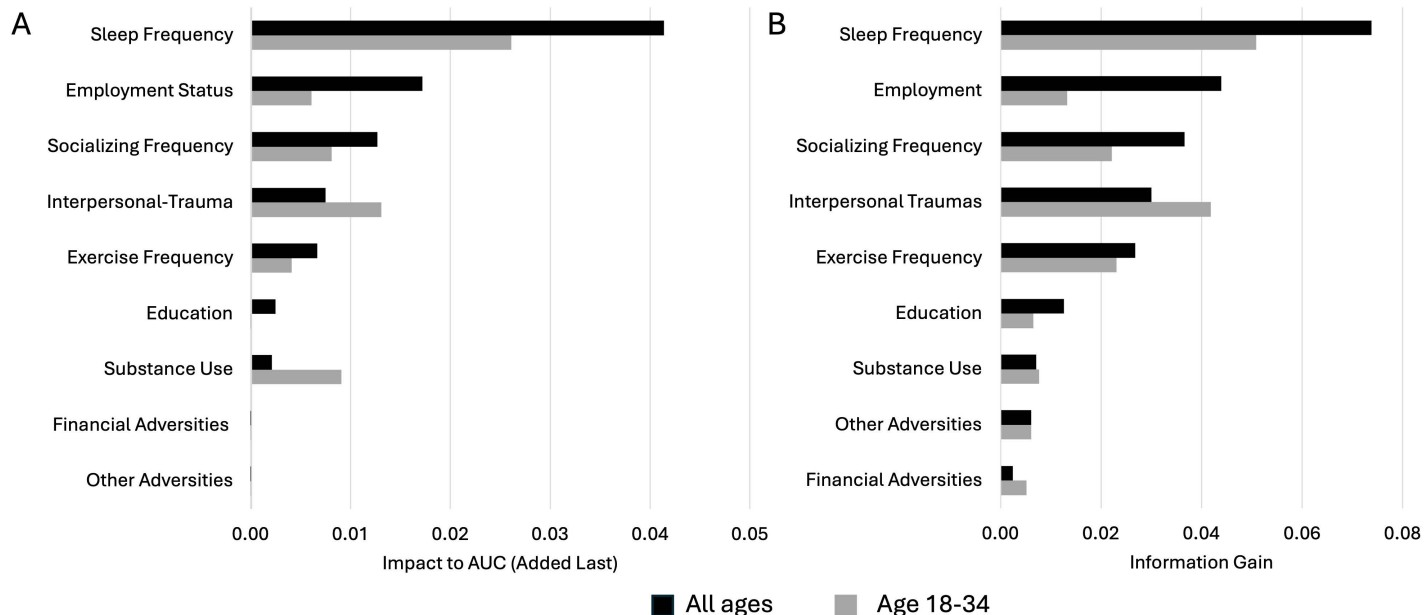

**Fig 3. Hierarchy of factors contributing to the classification of moderate to severe anxiety symptoms. (A)** Impact to AUC of adding the factor last after all other factors for all ages and 18 to 34. **(B)** Information Gain of factors. Legend spans both A and **B.**

**Table 4. Ranking of impact of factors on model performance (AUC and F1 scores) based on first or last inclusion in forward addition models.**

|  | Average Rank | | All ages | | | | 18-34 Age group | | | |
|---|---|---|---|---|---|---|---|---|---|---|
|  | All ages | 18-34 | AUC First | AUC Last | F1 First | F1 Last | AUC First | AUC Last | F1 First | F1 Last |
| Sleep | 1 | 1 | 1 | 1 | 2 | 1 | 1 | 1 | 1 | 1 |
| Employment | 3 | 6 | 2 | 2 | 6 | 3 | 7 | 6 | 7 | 5 |
| Social | **4** | **4** | 3 | 3 | 4 | 7 | 5 | 5 | 3 | 3 |
| Interpersonal-Trauma | 3 | 2 | 4 | 4 | 1 | 2 | 2 | 2 | 2 | 2 |
| Exercise | 5 | 4 | 5 | 4 | 3 | 7 | 4 | 4 | 4 | 2 |
| Substance Use | 6 | 6 | 6 | 6 | 8 | 4 | 6 | 7 | 8 | 4 |
| Education | 6 | 8 | 7 | 5 | 5 | 6 | 9 | 9 | 6 | 6 |
| Financial Adversities | 7 | 8 | 8 | 6 | 7 | 5 | 8 | 8 | 9 | 6 |
| Other Adversities | **7** | **4** | 9 | 6 | 7 | 5 | 3 | 3 | 5 | 6 |

Note: 'Impact' refers to the change in AUC or F1 score when each variable group is added in the forward-selection framework using the LASSO-regularized Logistic Regression classifier. Adding a factor 'first' (after the base model with biological sex) captures its contribution inclusive of correlations with other factors; adding 'last' reveals its independent contribution after accounting for shared variance. Actual delta values are provided in Table D in S1 Text R = Rank.

(average rank of adding first and last) for the 18–34 age group alone. However, employment status had little impact (ranked 6th) while the experience of interpersonal trauma ranked higher (2nd). Frequency of exercise and frequency of social interaction followed, jointly ranking 4th. The experience of other adversities (i.e., not financial or interpersonal, such as illness, injury, or natural disasters) also ranked 4th when added first, but had no impact when added last, similar to financial adversities and educational attainment.

## Hierarchy of factors using Information Gain

We similarly evaluated the hierarchy of factors associated with moderate to severe anxiety symptoms for all age groups together, and for younger ages 18–34 separately, using Information Gain, a model independent method (Fig 3B). Here again the results were consistent with the impact to the model as described above. In particular, the feature category of sleep dominated as the most important factor associated with moderate to severe anxiety symptoms across all age groups, followed by employment status, frequency of socializing and the experience of interpersonal trauma. Similarly, for the 18–24 age group alone, sleep was the most important factor, followed by the experience of interpersonal trauma, while employment status had a lesser impact.

The Information Gain values of each feature for all ages are shown in Table E in S1 Text. Getting a good night's sleep 'Hardly ever' or 'Most of the time' had the top two highest Information Gain values while 'Rarely/Never' exercising or socializing and being 'Retired' had the next highest. Bullying by peers and parental abuse or neglect in childhood as well as being 'Not able to work' or 'Unemployed', other frequencies of getting a good night's sleep and abuse or assault in childhood were also in the top 15 features.

## Factor contribution using SHAP

Finally, we used SHAP (SHapley Additive exPlanations) as a qualitative tool to highlight the directionality and consistency of associations, rather than as a definitive measure of causal importance (Fig 4). We show the direction of impact of the top 4 factors across all ages (frequency of getting a good night's sleep, employment status, frequency of socializing and frequency of exercising). It is important to note that when predictors are correlated, SHAP values share importance estimates across correlated predictors because they summarize contributions across all feature combination scenarios; however, the consistency of rankings across SHAP, Information Gain, and forward-selection analyses strengthens confidence in the identified hierarchy. Here, each individual was plotted as a point either in blue or red, where blue indicates that the option was not selected whereas red indicates it was selected. Values to the right of zero indicate how much it pushed the model towards a positive classification of moderate to severe anxiety symptoms while values to the left of zero indicate how much it pushed the model towards a negative classification. Selection of 'Hardly ever' having a good night's sleep consistently and substantially contributed towards a positive classification of moderate to severe anxiety symptoms, while having a good night's sleep only 'Some of the time' also contributed towards a positive classification, but to a lesser extent. In contrast, having a good night's sleep 'Most of the time' or 'All of the time' contributed to a negative classification. Similarly, 'Rarely/never' socializing in person contributed to a positive classification while socializing at least 1–3 times a month or more contributed to a positive classification. Finally, having an employment status of 'Not able to work' contributed most strongly to a positive classification of moderate to severe anxiety symptoms followed by a status of 'Homemaker' and 'Unemployed'. In contrast, being 'Employed', 'Retired' or 'Studying' contributed strongly to a negative classification.

## Discussion

Here we show that lifestyle and life context factors, particularly sleep quality, frequency of exercise, social interaction, and interpersonal trauma, play a substantial role in predicting moderate to severe anxiety symptoms, defined here as GAD-7 scores of 10 and above. The machine learning models employed here also revealed the hierarchy across these factors, with sleep quality being the most prominent across all age groups. Additionally, the influence of lifestyle factors on anxiety symptoms differs across age groups. These findings build on existing machine learning studies that, to date, have been smaller in scale [30–34] or scope [31,34–40,42–45] and are a first demonstration of the aggregate contribution of a large number of adversities and traumas together with lifestyle factors to the incidence of anxiety symptoms across a large-scale sample from the general population. Altogether, they highlight the complex interplay and hierarchy of factors associated with the experience of anxiety symptoms and have implications for targeted interventions and public health policies aimed at preventing and treating anxiety.

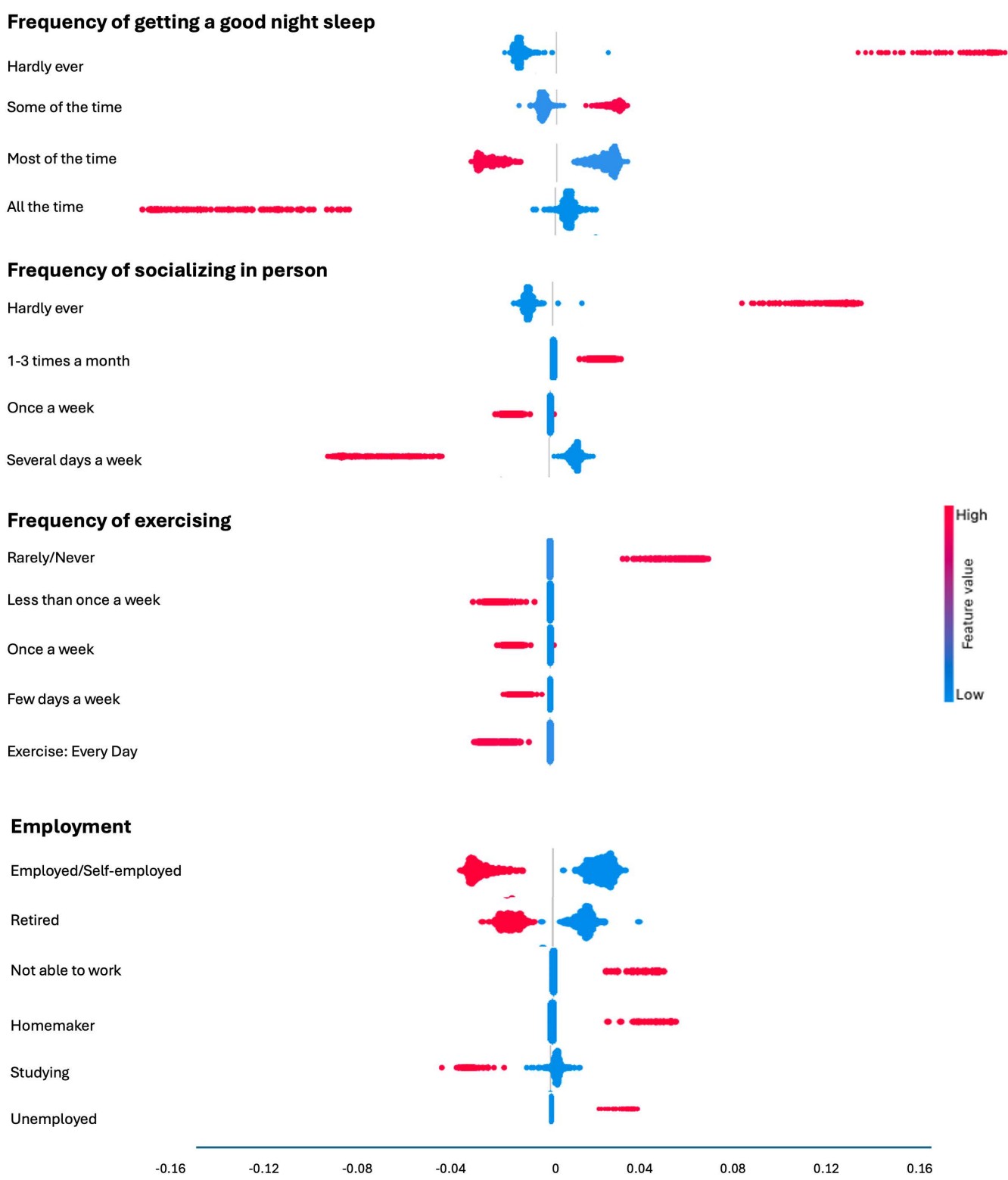

**Fig 4. SHAP values for each category of sleep frequency, socializing frequency and employment status.** A red dot represents individuals who selected the option, while a blue dot indicates individuals who did not select the option. Points to the right of 0 indicate a contribution to positive classification of moderate to severe anxiety symptoms, points to the left of 0 indicate a contribution to a negative classification.

## Model performance and differences by age

The lifestyle and life context factors used in this study had substantial predictive power for older age groups. However, their predictive power was systematically diminished for younger age groups. While this study included a broad range of adversities, traumas, and lifestyle factors, some factors that impact mental health outcomes such as diet [e.g., ultra-processed food, [55]] and social media use [56] were not included and could be substantial contributors in younger age groups. In addition, studies are increasingly showing an impact of environmental toxins on mental health [57,58] which could also play a role and remains to be studied. However, importantly, it suggests that the factors associated with symptoms of anxiety differ between older and younger generations, which has implications for how we approach both prevention and treatment. While this finding may appear counterintuitive given that adverse events are commonly reported among younger populations, the specific constellation of lifestyle and life context factors captured here—many representing cumulative life experiences such as employment history and interpersonal traumas—may be more salient risk markers for anxiety in older adults, whereas younger adults may be more affected by contemporary factors not included in this model. This pattern is consistent with our previous MHQ study [59], where model performance also improved systematically with age (accuracy 0.68 for ages 18–24 vs. 0.94 for ages 75–84). We also note that this generational trend is also true for the prediction of transdiagnostic mental distress [59].

## Hierarchy of life context factors driving anxiety

We used multiple methods to identify the hierarchy of life context factors associated with symptoms of moderate to severe anxiety. Across multiple methods we showed consistently that sleep status, and in particular frequent poor sleep, dominated the classification of moderate to severe anxiety symptoms for both age groups. For older age groups this was followed by employment status (for older ages), frequency of social interaction, number of interpersonal traumas, and frequency of exercise. In particular, for the older age group, not being able to work and being unemployed contributed most substantially to a classification of moderate to severe anxiety symptoms, while being employed or retired had the opposite effect. The experience of interpersonal trauma was a more predictive factor for the 18–34 age group, while employment status was a weaker factor. Among the individual interpersonal traumas, bullying by peers and abuse or neglect by a parent or caregiver in childhood were the strongest predictive factors.

However, although sleep quality was the most significant predictor of moderate to severe anxiety symptoms, even without the inclusion of this factor, other life factors such as social interaction, exercise, employment status and experience of interpersonal trauma, in aggregate, predicted moderate or severe anxiety symptoms with an AUC of 0.75 and F1 score of 0.73. This suggests firstly, that anxiety symptoms are predictably associated with people's lifestyle habits (i.e., rarely/never socializing or exercising) and certain types of adverse life experiences (i.e., life challenges such as not being able to work, unemployment, and the experience of various types of abuse or assault) and secondly, that these factors likely contribute to poor sleep with reciprocal feedback [60–62]. Although this study was cross-sectional in design and therefore cannot categorically distinguish between causality and consequence of symptoms, these findings point to the substantial sociological basis of anxiety symptoms that could be prevented and mitigated through shifts in culture and economics.

## Similarity of factors to transdiagnostic predictions

Anxiety is substantially comorbid with various disorders such as depression, obsessive-compulsive disorders (OCD) and panic disorder [63,64] among others. In line with this, we have previously shown that the same set of lifestyle and life context factors were similarly able to predict overall mental distress [59], as measured by the MHQ, a transdiagnostic measure that aggregates across 47 symptoms spanning 10 disorders [65]. That study used a larger sample (N = 270,000) collected between April 2020 and December 2021, whereas the present study uses a distinct sample (N = 4,186) collected in September 2022 with the GAD-7 as the outcome measure. The comparison between disorder-specific (GAD-7) and

transdiagnostic (MHQ) prediction provides unique insight into anxiety-specific risk factors. While the top 5 categories of factors were the same across both studies, there are certain differences worth noting. In particular, social interaction was the top contributor to prediction of the MHQ, followed by sleep quality, exercise frequency, employment status and experience of interpersonal trauma. In contrast, sleep quality played a more dominant role in the prediction of moderate to severe anxiety symptoms. This suggests that the specific symptoms of anxiety may be more tied to sleep than other aspects of mental distress.

**Strengths and limitations**

Key strengths of this study include the wide range of variables studied which integrate lifestyle habits with adverse experiences; the use of multiple algorithms including hierarchical analysis; and the ability to stratify the findings by demographics such as age and geography. However, there are several limitations to note. First, the study is a cross-sectional and therefore cannot distinguish between causality and consequence, especially given the bi-directional nature of many of the factors investigated here. However, this multi-variate data provides a unique opportunity to examine the hierarchy of impact of a broad range of factors on anxiety in a cost-effective and timely fashion and can provide well-evidenced hypotheses for interventional testing. Second, although a wide number of lifestyle and life context factors were used, several key factors, were missing. These factors are now included in more recent iterations of the MHQ and will be included in future analyses. Third, the sample population included in this study, while large-scale and obtained through tailored outreach, was a non-probability sample that may not be representative of the general population. As the assessment is performed online, this is particularly the case in countries where internet penetration is lower. Additionally, all measures were self-reported, which may be subject to recall bias and social desirability effects. Furthermore, while machine learning techniques offer advantages in handling complex, multivariate relationships, they are susceptible to overfitting and may not generalize well to populations with different characteristics from the training sample. However, comparisons of the Global Mind Data from the US showed that it is broadly comparable to data from the US Census. Studies are underway to perform similar comparisons for other countries where equivalent national statistics are available.

**Implications for approaches to prevention and treatment**

Given the known bidirectional association between anxiety and life context, these findings have implications for how we approach both prevention and treatment of anxiety. From a treatment perspective, it suggests that it is important to first determine an individual's specific life context before making treatment decisions relating to anxiety. At an individual level, lifestyle factors such as exercise and regular social interaction could substantially reduce the risk of severe anxiety symptoms evoked by adversity. In addition, a better understanding of sleep mechanisms and targeting underlying problems of sleep challenges could also be a possible path to treatment of anxiety symptoms, particularly when found to occur in the absence of obvious adverse events or lifestyle risk factors. For instance, sleep apnea or poor sleep hygiene may drive sleep challenges that in turn cause greater anxiety. Finally, at a population level, age-tailored socioeconomic programs aimed at increasing employment opportunities and reducing interpersonal abuse and assault could substantially decrease the incidence of anxiety.

## Supporting information

**S1 Text. Table A.** Percentage of respondents by age and biological sex. **Table B.** Percentage of respondents by country. **Table C.** Model results for all model types. **Table D.** Ranking of impact of factors on model performance (AUC and F1 scores) based on first or last inclusion in forward addition models. **Table E.** InfoGain values of each answer option for all ages.
(DOCX)

## Acknowledgments

We thank members of the Sapien Labs team for their assistance with recruitment and data management. We are grateful to all survey respondents for their participation in the Global Mind Project.

## Author contributions

**Conceptualization:** Jerzy Bala, Jennifer J. Newson, Tara C. Thiagarajan.

**Data curation:** Jerzy Bala.

**Formal analysis:** Jerzy Bala.

**Investigation:** Tara C. Thiagarajan.

**Methodology:** Jerzy Bala, Jennifer J. Newson, Tara C. Thiagarajan.

**Project administration:** Jennifer J. Newson.

**Supervision:** Tara C. Thiagarajan.

**Writing – original draft:** Jerzy Bala, Jennifer J. Newson, Tara C. Thiagarajan.

**Writing – review & editing:** Jerzy Bala, Jennifer J. Newson, Tara C. Thiagarajan.

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
