## [Decision Letter · Decision Letter 0]

10 Nov 2025

PMEN-D-25-00320

Contribution of social determinants to symptoms of generalized anxiety disorder

PLOS Mental Health

Dear Dr. Thiagarajan,

Thank you for submitting your manuscript to PLOS Mental Health. I am sorry for the delay. After careful consideration of the reviewer reports, we feel that your paper has merit but does not fully meet PLOS Mental Health’s publication criteria as it currently stands. Therefore, we invite you to submit a revised version of the manuscript that addresses the points raised during the review process.

Please address all of the comments raised, which you can find below. You will notice that Reviewer 1 was especially concerned about the rationale and analysis approach and so it is important that you focus on these particular aspects. We will be unable to proceed with the paper if these concerns are not fully addressed.

We look forward to receiving your revised manuscript.

Kind regards,

Karli Montague-Cardoso

Staff Editor

PLOS Mental Health

Journal Requirements:

1. We noticed you have some minor occurrence of overlapping text with the following previous publication(s), which needs to be addressed:

- https://doi.org/10.3389/fpsyt.2025.1576964

- https://doi.org/10.1136/bmjopen-2023-075095

In your revision ensure you cite all your sources (including your own works), and quote or rephrase any duplicated text outside the methods section. Further consideration is dependent on these concerns being addressed.

2.  Please note that PLOS Mental Health has specific guidelines on code sharing for submissions in which author-generated code underpins the findings in the manuscript. In these cases, we expect all author-generated code to be made available without restrictions upon publication of the work. Please review our guidelines at https://journals.plos.org/mentalhealth/s/materials-and-software-sharing#loc-sharing-code and ensure that your code is shared in a way that follows best practice and facilitates reproducibility and reuse.

3. Please amend your detailed Financial Disclosure statement. This is published with the article. It must therefore be completed in full sentences and contain the exact wording you wish to be published.

i. Please clarify all sources of financial support for your study. List the grants, grant numbers, and organizations that funded your study, including funding received from your institution. Please note that suppliers of material support, including research materials, should be recognized in the Acknowledgements section rather than in the Financial Disclosure.

ii. State the initials, alongside each funding source, of each author to receive each grant. For example: "This work was supported by the National Institutes of Health (####### to AM; ###### to CJ) and the National Science Foundation (###### to AM)."

iii. State what role the funders took in the study. If the funders had no role in your study, please state: “The funders had no role in study design, data collection and analysis, decision to publish, or preparation of the manuscript.”

iv. If any authors received a salary from any of your funders, please state which authors and which funders.

4. Please ensure that your Ethics Statement is available in its entirety at the beginning of your Methods section, under a subheading 'Ethics Statement'. It must include:

1) The name(s) of the Institutional Review Board(s) or Ethics Committee(s)

2) The approval number(s), or a statement that approval was granted by the named board(s)

3) (for human participants/donors) - A statement that formal consent was obtained (must state whether verbal/written) OR the reason consent was not obtained (e.g. anonymity).

5. Please provide separate figure files in .tif or .eps format.

https://journals.plos.org/mentalhealth/s/figures

https://journals.plos.org/mentalhealth/s/figures#loc-file-requirements

6. For studies involving third-party data, we encourage authors to share any data specific to their analyses that they can legally distribute. PLOS recognizes, however, that authors may be using third-party data they do not have the rights to share. When third-party data cannot be publicly shared, authors must provide all information necessary for interested researchers to apply to gain access to the data. (https://journals.plos.org/plosone/s/data-availability#loc-acceptable-data-access-restrictions

Additional Editor Comments (if provided):

Reviewers' comments:

Reviewer's Responses to Questions

**Comments to the Author**

1. Does this manuscript meet PLOS Mental Health’s publication criteria?

Reviewer #1: No

Reviewer #2: Yes

2. Has the statistical analysis been performed appropriately and rigorously?

Reviewer #1: No

Reviewer #2: Yes

3. Have the authors made all data underlying the findings in their manuscript fully available (please refer to the Data Availability Statement at the start of the manuscript PDF file)?

Reviewer #1: Yes

Reviewer #2: Yes

4. Is the manuscript presented in an intelligible fashion and written in standard English?

Reviewer #1: Yes

Reviewer #2: Yes

Reviewer #1: The abstract gives me the impression that the results will be a hodge-podge, as the types of predictors listed include potential causes (e.g., childhood adversities), prodromes (e.g., sleep problems), and endogenous variables (e.g., social isolation). And the classifiers listed included most that are predictive and one, SHAP, that merely helps interpret results from the others. We are told nothing in the abstract about sample size, design, or strength of associations.

Line 52 Specific phobia and adjustment disorder are much more common than GAD

Line 89 Sentence structure

Line 95 You’re not really “predicting” if the study is cross-sectional and some of the variables are likely to have bidirectional associations with GAD.

Lines 122-2 I don’t understand the rationale for this exclusion based on standard deviation. On the face of it, it does not make sense. Spell out the rationale.

Line 132 Based on absence of mention, I take it that no calibration was used to make this sample representative of the population on known population characteristics.

Line 150 I don’t get this. You estimated a model for having GAD and another model for not having GAD? Why? This was to deal with missing outcomes data?

The description of the analysis approach didn’t make much sense. Nor did the presentation of results. We’d normally expect lasso to be the benchmark model and to see how much you could improve on thi with more complex models. But I never saw a table reporting this. And Table 4 doesn’t tell us either classifier on which results are based on what the meaning of “impact” is. Are these rank orders of SHAP Values? From what model?

It's important to appreciate that SHAP values do not tell you unequivocally about predictor importance. When predictors are correlated, estimates of importance get shared across correlated predictors because the SHAP value summarizes across all scenarios, in some on which those correlated predictors are present. Your interpretations are naïve.

Reviewer #2: This is a very interesting study, and the authors should be commended for a well-written and impactful manuscript. Specific comments follow:

1. Given that the title mentions the contribution of social determinants, the term itself does not appear in the manuscript. It is unclear whether the authors are referring to social determinants of health or specifically of mental health. The study explores contextual factors such as life trauma, poor sleep quality, infrequent exercise, unemployment, and social isolation, so it would be helpful to clarify why the term “social determinants” was not used.

2. While the variables are outlined on pages 4 and 5, there is limited information on how demographics, geography, lifestyle, substance use, medical history, and childhood interpersonal trauma were measured. For example, it is clear that GAD-7 was used to assess anxiety symptoms, but it is not clear how the other measures were defined and operationalised.

3. On page 8, the manuscript states that only respondents who answered “yes” to the MHQ question, completed the survey in a timeframe appropriate for reading all questions (above 7 minutes), and had a standard deviation above 0.2 across MHQ-rated responses were included in the analysis. The authors should clarify why these criteria were chosen and how they were assessed.

4. On page 11, the term “imbalanced datasets” is introduced without explanation. It is unclear what this refers to, for example, was the dataset imbalanced due to unequal class distributions, missing data or another reason? Additionally, on page 12, it is noted that logistic regression did not achieve the highest scores in any single metric. It would be helpful to briefly explain why this might be the case. The term “geographies” is also introduced on page 12 in the statement “regression models combining all geographies were used for further analysis.” The authors should clarify what “geographies” refers to in this context. Further, how was missing data handled?

5. The finding that older age groups were more likely to experience anxiety symptoms associated with lifestyle and adverse life circumstances, while younger age groups were increasingly affected by other factors not included in the model, is interesting. However, this seems counterintuitive, as anxiety symptoms associated with adverse events are often more prevalent among younger populations. The authors may wish to elaborate on this finding, if necessary.

6. In the limitations section, the authors could consider noting the use of self-reported measures, and the potential limitations of machine learning techniques and algorithms.

**Do you want your identity to be public for this peer review?** For information about this choice, including consent withdrawal, please see our Privacy Policy

Reviewer #1: No

Reviewer #2: No

---

## [Editor Report · Decision Letter 1]

19 Jan 2026

Contribution of social determinants to symptoms of generalized anxiety disorder

PMEN-D-25-00320R1

Dear Dr. Thiagarajan,

We are pleased to inform you that your manuscript 'Contribution of social determinants to symptoms of generalized anxiety disorder' has been provisionally accepted for publication in PLOS Mental Health.

Best regards,

Karli Montague-Cardoso

Staff Editor

PLOS Mental Health